# Numerical Prediction of Strength of Socket Welded Pipes Taking into Account Computer Simulated Welding Stresses and Deformations

**DOI:** 10.3390/ma15093243

**Published:** 2022-04-30

**Authors:** Tomasz Domański, Wiesława Piekarska, Zbigniew Saternus, Marcin Kubiak, Sebastian Stano

**Affiliations:** 1Faculty of Mechanical Engineering and Computer Science, Czestochowa University of Technology, Dabrowskiego 69, 42-201 Czestochowa, Poland; zbigniew.saternus@pcz.pl (Z.S.); marcin.kubiak@pcz.pl (M.K.); 2Faculty of Architecture, University of Technology, Civil Engineering and Applied Arts, Rolna 43, 40-555 Katowice, Poland; wieslawa.piekarska@pcz.pl; 3Łukasiewicz Research Network-Institute of Welding, Błogosławionego Czesława 16-18, 44-100 Gliwice, Poland; sebastian.stano@is.lukasiewicz.gov.pl

**Keywords:** socket welding, image correlation system, numerical analysis, thermomechanical phenomena, deformations, welded pipes

## Abstract

The paper presents a numerical model based on the finite element method (FEM) to predict deformations and residual stresses in socket welding of different diameter stainless steel pipes made of X5CrNi18-10 steel. The next part of the paper concerns the determination of strength properties of a welded joint in terms of a shear test. A thermo-elastic–plastic numerical model is developed using Abaqus FEA software in order to determine the thermal and mechanical phenomena of the welded joint. This approach requires the implementation of moveable heat source power intensity distribution based on circumferentially moving Goldak’s heat source model. This model is implemented in the additional DFLUX subroutine, written in Fortran programming language. The correctness of the assumed model of thermal phenomena is confirmed by examinations of the shape and size of the melted zone. The strength of the welded joint subjected to shear is verified by performing a compression test of welded pipes as well as computer simulations with validation of the computational model using the Dantec 3D image correlation system.

## 1. Introduction

Welding is an integral technological process performed during the production of many structural elements. This process has a direct impact on the integrity as well as thermal and mechanical behavior of construction in operational conditions. Welded constructions are an essential part of many buildings, bridges, ships, pipes, pressure vessels, nuclear reactors and other engineering structures [1,2,3,4]. 

Undoubtedly one of the most advantageous sections commonly used in steel construction is pipe sections. They are used as elements for the transmission of fluids and gases (pipelines as well as various types of supporting of structures and trusses). This is because they provide maximum buckling strength (greatest moment of inertia) with a minimum use of material in the construction [5,6]. In the case of pipelines, two basic methods of joining are used—butt welding and socket welding [7,8]. 

Used in pipelines, socket joints of the same diameter require expanding the cross-section of one end of pipe segment to the other end of the adjacent pipe segment which is inserted. On the other hand, lap joints for pipes with different diameters (used as pipe reducer or in long constructions with enlarged stiffness) require contact fitting joined pipes or a gentle undercut of the inner diameter of a wider pipe segment to fit with the narrower pipe segment [9,10]. 

Butt-welded pipes are characterized by low construction costs. On the other hand, the weld zone of the socket-welded pipe is a weak region compared with the butt weld zone, especially in the case of fatigue strength [11,12,13]. Properly performed butt welding ensures optimal static and fatigue strength of a join. However, it requires great skills of the welder and large expenses related to the preparation of pipes before welding. Socket welding of pipes is often used as an alternative to butt-welding or welding between the pipe and fitting such as valve, union, tee, orifice, or elbow. 

This type of joint is used especially in the case when butt joints or flange joints cannot be made or when construction requires weight reduction with maintained desired stiffness (e.g., in street lighting) or as repair joints on existing pipelines. Moreover, socked welding is used for small-bore pipes (with a nominal diameter less than 50.8 mm/2 inches) in secondary piping systems in nuclear power plants, with a special condition that they should not be used in service where crevice corrosion between the pipe and fitting may occur. There are usually about 40,000 socket welds in one typical 1000 M pressurized water reactor (PWR) plant [9,10,11]. 

Mechanical parameters are essential for the proper operation of the circumferentially welded tubular structures. A large amount of heat introduced into the joint has a significant impact on its strength properties [14,15,16,17] of welded pipes. Recognition of values of residual stresses is extremely important when analyzing the development of cracks in welded constructions [18,19]. Their evaluation can help in solving problems related to intercrystalline of stress fracture or fatigue strength. Moreover, it is important to determine the deformation capacity of steel pipelines, especially in pipelines constructed in geohazard areas (e.g., areas with seismic activity) [20,21]. 

Numerical prediction of the thermo-mechanical properties of welded joints and the selection of welding parameters can significantly accelerate the implementation and reduce the costs of the technological process. Over the past decade, a number of numerical models have been developed to evaluate the temperature distribution and residual stresses for welding of steel pipes [5,6,7,8,16,17,18,22]. Researchers use a full three-dimensional numerical model [8,18,23,24,25] to analyze the effect of changing parameters during circumferential welding of pipes on the distribution of temporary and residual stresses, to analyze stress state resulting from joining dissimilar materials or to simulate the residual stresses during multi-pass welding of a pipe Many researchers choose axisymmetric 2D models to reduce computation time and costs in simulations of circumferential welding of pipes [4,5,26,27]. Most numerical models are verified by the real welding tests and strength test made on appropriate strength testing machines [19,28,29,30]. There is still a lack of verification of numerical models on the basis of experimentally determined field of values (such as displacement field) in the entire area of the analyzed sample. The current research in the field of numerical modelling of pipe welding is focused mainly on the analysis of thermomechanical phenomena in butt joints [8,14,15,16,17]. There are only few papers available in the literature concerning numerical analysis of thermomechanical phenomena occurring in socket-welded pipes.

This work presents the physical aspects of welding with a concentrated, moveable heat source around the outer surface of pipes and numerical analysis of compression of welded pipes having different diameters. The computations are made in Abaqus FEA calculation software, extended by additional numerical subroutines. Thermomechanical properties of steel X5CrNi18-10 varying with temperature are adapted in calculations [16,31,32,33]. On the basis of [8,10,18,19,20,25] the energy parameters of the process are verified. The numerical model is based on the geometry of experimental samples of pipes with different diameters, circumferentially lap welded using GTAW method. Simulations were performed to determine temperature field in the joint, the shape and size of the melted zone and stress state as well as displacement field. In order to verify the correctness of the adopted heat transfer models, the obtained shape and size of the melted zone is compared with macroscopic picture of the cross-section of the joint. Longitudinal loads are the main cause of the destruction of socket-welded joints. The scope of the research concerning the numerical modelling of socket-welded joints should also include a strength analysis. The compression test of the circumferentially welded pipes presented in the paper is indeed a shear test of the welded joints. Welded pipes were compressed using Zwick/Roell universal testing machine in order to determine the strength of the welded joint during compression. Dantec 3D image correlation system was used in the compression test to measure the real displacement distribution in various zones of welded pipes. The obtained results of numerical simulations of the compression test were compared with the results of experimental tests.

## 2. Experiment

During the GTAW welding experiments, two pipes of different diameters made of austenitic steel were circumferentially lap welded. Dimensions are: inner pipe with diameter of 30 mm × 2 mm and length 66.5 mm and outer pipe with a diameter 33.7 mm × 2 mm and length 65.5 mm. Diagram of the system and the dimensions are shown in Figure 1. 

The pipes were joined with 20 mm overlap. The outer tube was rolled up to 0.15 mm over a length of 20 mm for a good assembly of the joint (Figure 1). The welding process was performed with the use of an additional material in the form of a rod with a diameter of 1 mm. Argon gas was used as a shielding gas. The welding process parameters were: current 83 A, voltage 20 V, speed of the torch 0.3 m/min, the angle of deflection of the welding torch from the vertical plane is 20°. Figure 2 shows obtained welded joint. 

The metallographic tests were performed to determine the size and shape of the melted zone. The data obtained from experimental tests are necessary in the verification of the heat source power distribution model. The macroscopic picture of welded joint allowed comparing the shape of the melted zone with the results of the numerical simulations. Accurate determination of the heat load ensured proper temperature distribution in the joint and appropriate analysis of welding stresses and deformations [19,24]. 

## 3. Image Correlation System

Measuring methods using image correlation are now more and more often used to determine the components of stresses, strains or displacements in laboratory conditions as well as to identify defects in construction elements or machines generated during static or dynamic loads [34,35,36,37,38]. The correlation algorithm tracks the position of the same points in the source image and the distorted image (Figure 3). 

Correlation algorithms make it possible to determine the maximum displacement with an accuracy of 1/100 of a pixel of the matrix. The correlation algorithm tracks the position of the same points visible in the source image and the distorted image. To achieve this, a square surface containing a set of pixels is identified in the source image and at a position appropriate for the image after deformation. 

Figure 4 shows a diagram of the research station, which distinguishes three basic groups of elements: the tested object (sample with the loading system), the measuring system and the system analyzing results of the measurement. The measuring system is equipped with a set of digital cameras mounted on a common tripod. This system is coupled with a computer unit equipped with ISTRA 3D software, which collects and analyzes results recorded during the measurement.

### 3.1. Measurement System

The research uses a universal testing machine Zwick & Roell Z100 with maximum load 100 kN and precision 1 N force/0.01 mm displacement (without extensometer, Figure 5). The measurement system is connected with Image Correlation System Dantec 3D. Dantec system is equipped with optical cameras used to record strains in the tension test with 50 mm lenses and have maximum resolution 2048 × 2048 pixels each. This allowed determining the full size of analyzed sample in the working area of the universal testing machine. 

A system of three cameras was used in the experiment. Strain fields were measured for the entire tension cycle. A trigger mechanism was created in Istra4D software for the measurement. Pictures were made for every time increment Δt = 0.1 s. 

### 3.2. Results of Measurements 

Experimental tests were performed in order to verify the correctness of the numerical model [39,40] of thermomechanical phenomena involving the compression of welded pipes up to 90 kN (Figure 6a). Figure 6b shows a diagram of the displacement of the upper traverse of the testing machine as a function of the duration of load. The compression process was recorded by a system of cameras. Cameras took 471 control photos during the test. The load was applied after reaching the initial force equal to 100 N. The unloading process took place after exceeding the maximum force of 90 kN. Figure 7 shows the displacement values at the selected five control points along *z* axis. Figure 7 contains the full characteristics of the compression test with separate zones: basic load and unloading. The loading time starts from step 0 and ends at step 375. Characteristic control point for which the maximum displacement value obtained from the movement of the traverse corresponds to the value measured by the Dantec system are pointed out in Figure 6 and Figure 7. 

Figure 8a shows the displacement field along the *z* axis (the axis compliant with the loading direction), where the measurement line is marked as a solid line. For this line, the displacement diagram is shown in Figure 8b. The measuring line is on the side surface of welded pipes. It can be observed that in the area of the weld; the displacements have much lower values compared to the base material, which is related to the increase in stiffness of this area, resulting from the change of mechanical properties of the weld. 

The displacement distribution *U_x_*, *U_y_* over time 37.5 s (which corresponds to the action of specific load) on the side surface of the pipes is shown in Figure 9. The visible decrease in the value of the displacement occurs within the weld in the perpendicular direction to the side surface (axis *y*). A slight increase in the displacements along the *x* axis (horizontal) is caused by a small deflection of the sample caused by a low slip of the pressure plate.

## 4. Mathematical and Numerical Modeling of Circumferential Welding of Pipes

### 4.1. Mathematical Models of Thermomechanical Phenomena

Numerical analysis of thermomechanical phenomena is divided into thermal and mechanical analysis [19,41,42]. 

Thermal phenomena are described by a heat transfer equation in the Abaqus/Standard simulation module. The solution equation is based on the law of energy conservation and Fourier’s law [27]. This equation is described by the following formula: (1)$$\int_{V}\rho \frac{\partial U}{\partial t}\delta TdV+\int_{V}\frac{\partial \delta T}{\partial x_{\alpha }}\cdot \left(\lambda \frac{\partial T}{\partial x_{\alpha }}\right)dV==\int_{V}\delta T\;q_{V}dV+\int_{S}\delta T\;q_{S}dS$$
where λ = λ(*T*) is thermal conductivity (W/(m K)), *U = U*(*T*) is internal energy (J/kg), *δT* is variational function, *ρ* is density (kg/m^3^), *q_v_* is volumetric heat source (W/m^3^), and *q_S_* is a surface heat flux (W/m^2^).

Equation (1) is completed by the initial condition and boundary conditions of Dirichlet, Neumann, and Newton type [19,33]. 

Numerical analysis of mechanical phenomena in Abaqus FEA is based on classical equilibrium equations. Thermal deformations dominate in the analyzed system; therefore, small deformations are assumed in the analysis [43,44,45]. The displacement gradient is small; hence, the deformation tensor can be written as:(2)$$\varepsilon =\frac{1}{2}\left[\nabla u+{\left(\nabla u\right)}^{T}\right]\;when\;\nabla u=\frac{\partial u}{\partial u}\approx 1$$

Cauchy stress tensor is identified with the nominal stress tensor, according to the following equation:(3)$$\nabla \circ \sigma \left(x\right)+F_{b}=0\;$$
where **σ** is Cauchy stress tensor, ∇ is divergence operator, **F_b_** is the body force density with respect to initial configuration.

The above equilibrium equation is completed by boundary conditions, given as: (4)$$\begin{array}{l} \;u=\hat{u}\;for\;{\;S}_{u}\\ \sigma \cdot n=\Phi \;for\;S_{\Phi } \end{array}$$
where **û** is displacement on boundary surface portion **S_u_** and **Φ** is boundary surface tractions on portion **S_Φ_**.

Cauchy stress tensor, ∇, is divergence operator, **F_b_** is the body force density with respect to initial configuration. Total strain is described using elastic–iscoplastic model according to [19,43,44]:(5)$$\varepsilon =\varepsilon^{total}=\varepsilon^{e}+\varepsilon^{p}+\varepsilon^{Th}$$
where *ε^e^* is elastic strain rate tensor, *ε^p^* is plastic strain rate tensor and *ε^Th^* is thermal strain rate.

Stress rate depends on elastic strain rate
(6)$$\dot{\sigma }=\underset{=}{D}:\left(\dot{\varepsilon }-{\dot{\varepsilon }}^{p}-{\dot{\varepsilon }}^{Th}\right)$$
where $\underset{=}{D}$ is the fourth order isotropic elasticity tensor, “:” is inner tensor product.
(7)$$\underset{=}{D}=2\mu \underset{=}{I}+\left(k_{B}-\frac{2}{3}\mu \right)I⊗I$$
where *μ*, *k_B_* is the shear and bulk modulus, $\underset{=}{I}$, I is fourth and second order identity tensors and ⊗ is outer tensor product. 

Elastic strain is calculated using inverted generalized Hook’s law, described by the formulation [43,44]: (8)$$\varepsilon_{ij}^{e}=\frac{1}{E}\left[\left(1+\upsilon \right)\sigma_{ij}-\upsilon \sigma_{kk}\delta_{ij}\right]$$
where *E* is Young’s modulus, *ν* indicates Poisson’s ratio and *σ_ij_* is stress tensor and *δ_ij_* is Kronecker delta. 

Plastic strains are determined on the basis of the model of non-isothermal plastic flow with the Huber–Mises plasticity condition and isotropic strengthening. The flow function (*f*) is determined according to the following equation:(9)$$f=\sigma_{ef}-\bar{\sigma }\left(\varepsilon_{ij}^{p},T\right)=0$$
where $\sigma_{ef}$ is effective stress, $\bar{\sigma }\left(\varepsilon_{ij}^{p},T\right)$ is material plasticizing stress—dependent on plastic deformation ($\varepsilon_{ij}^{p}$) and temperature *T*. 

Effective stress and strain are described as follows: (10)$$\sigma_{ef}=\sqrt{\frac{3}{2}S_{ij}S_{ij}}\;and\;{\dot{\varepsilon }}_{ef}^{p}=\sqrt{\frac{2}{3}{\dot{\varepsilon }}_{ij}^{p}{\dot{\varepsilon }}_{ij}^{p}}$$
where *S_ij_* is a deviatoric stress tensor $\left(S_{ij}=\sigma_{ij}-\frac{1}{3}\sigma_{m}\delta_{ij}\right)$, $\sigma_{m}$ is an average stress.

The plastic deformation rate can be expressed in the following form
(11)$${\dot{\varepsilon }}_{ij}^{p}=\frac{3}{2}{\dot{\varepsilon }}_{ef}^{p}\frac{S_{ij}}{\sigma_{ef}}$$
where ${\dot{\varepsilon }}_{ij}^{pl}$ is plastic strain rate component, λ signifies the plastic flow factor and *S_ij_* represents the deviatoric stress.

Thermal strain occurs as a result of changes in volume due to temperature differences:(12)$$\varepsilon_{ij}^{Th}=\int_{T_{0}}^{T}\alpha (T)dT\delta_{ij}$$
where α is the temperature-dependent coefficient of thermal expansion, *T*_0_ is the reference temperature.

### 4.2. Modelling of the Heat Source

In the case of numerical modeling of the electric arc-welding process, the Goldak model is most often used in the literature to describe the source power distribution [33,46,47]. This model is described by two half ellipsoids connected together by a symmetry axis [46]. The model diagram is shown in Figure 10. The equation describing Goldak’s model is expressed in the following equation:(13)$$\begin{array}{l} Q_{1}(x,y,z)=\frac{6\sqrt{3}f_{1}Q_{A}}{abc_{1}\pi \sqrt{\pi }}\exp(-3\frac{x^{2}}{c_{1}{}^{2}})\exp(-3\frac{z^{2}}{a^{2}})\exp(-3\frac{y^{2}}{b^{2}})\\ Q_{2}(x,y,z)=\frac{6\sqrt{3}f_{2}Q_{A}}{abc_{2}\pi \sqrt{\pi }}\exp(-3\frac{x^{2}}{c_{2}{}^{2}})\exp(-3\frac{z^{2}}{a^{2}})\exp(-3\frac{y^{2}}{b^{2}}) \end{array}$$
where parameters *a*, *b*, *c*_1_ and *c*_2_ are described dimensions of shape’s Goldak heat source, coefficients *f*_1_ representing energy distribution in the front of the heat source and coefficients *f*_2_ representing energy distribution in the back of the heat source, (*f*_1_ + *f*_2_ = 2), satisfying the condition *Q*_1_(*x*,*y*,*z*) and *Q*_2_(*x*,*y*,*z*).

In Equation (13) parameter *Q_A_* describes the value of the electric arc power: (14)$$Q_{A}\;=\;I\cdot U\cdot \eta$$
where *U* is voltage [V], *I* is current intensity [A] and *η* is the efficiency. 

Additional subroutine DEFLUX is implemented into Abaqus FEA solver to define a moveable welding source [19,33]. The subroutine includes a mathematical model of the distribution of heat source power, speed and direction of source travel. The main aspect of modelling the circumferential welding process in Lagrange coordinates is the analysis of the source motion along the circle path (Figure 11).

It is necessary to introduce transformation equations in order to reflect the position of the heat source in Cartesian coordinates. The equations of transition from the cylindrical system to the Cartesian system are introduced into the DFLUX procedure. Transition equations are described below:(15)$$\left\{ \begin{array}{l} x=R_{Z}\sin(\phi_{0}+\omega \cdot t)\\ y=R_{Z}\cos(\phi_{0}+\omega \cdot t)\\ z=z_{0} \end{array}\right.$$
where *R_z_* is an outer pipe radius, *t* is a time, $\phi_{0}$ is initial position the axis of the beam, $\omega =v/R_{z}$ is the angular speed, in which *v* = const. is a linear speed along the perimeter of the pipe, and *z*_0_ is the initial position on *z*-axis. 

In order to include the source inclination in the model, it is necessary to introduce an additional transformation with respect to the x axis to the calculations. Figure 12 shows a schematic representation of transformations. 

The transformation of welding source power distribution is carried out using transformation model [33]: (16)$$A_{i^{\prime }}=\gamma_{i^{\prime }j}\;A_{j}\;where\;\gamma_{i^{\prime }j}=\underset{i^{\prime }}{e}\;\cdot \underset{j}{e}$$
(17)$$\left\{ \begin{array}{l} x=x^{\prime }\\ y=-\cos\;\alpha \;\cdot y^{\prime }-\sin\;\alpha \;\cdot z^{\prime }\\ z=\sin\alpha \;\cdot y^{\prime }-\cos\alpha \;\cdot z^{\prime } \end{array}\right.$$

Equation (16) is obtained after the solution of presented transformation matrix from the basic system (*x*, *y*, *z*) to the rotated system (*x*_1_, *y*_1_, *z*_1_). Transformation equations have also been written in the DFLUX subroutine. Numerical modelling of circumferential welding of two pipes of different diameters also requires the appropriate angle of inclination of the heat source axis. Figure 13 shows the inclination of heat source adopted in the model. The axis of the welding source is rotated by an angle α = 20° (the inclination direction is the same as in the experiment).

### 4.3. Numerical Model

The numerical model in Abaqus FEA is developed using the real welded pipes (shown in Figure 1). The model of geometry of the welded pipes is presented in Figure 14 with the finite element mesh. 

The smaller FE-mesh step is about 0.25 mm and it occurs in the heat source activity zone. The total number of finite elements is 495,100. The material model of steel X5CrNi18-10 is assumed from literature data [32,33] as: solidus and liquidus temperatures *T*_S_ = 1400 °C, *T*_L_ = 1455 °C, latent heat of fusion H_L_ = 260 × 10^3^ J/kg, ambient temperature *T*_0_ = 20 °C and the heat convection *α*_k_ = 50 W/m^2^. 

The basic parameters of Goldak’s source are assumed using data from the experiment. The power of the heat source is determined on the basis of Equation (13) with Q_A_ = 996 W (assuming the efficiency of the process *η* = 60%). Welding speed is set to *v* = 0.3 [m/min], and the angle of deflection of the welding torch from the vertical plane is set to 20°. On the basis of numerical verification, the following parameters of Goldak’s source are assumed: a = 4 mm, b = 2 mm, c_1_ = 4 mm and c_2_ = 4 mm. Coefficients are *f*_1_ = 1 and *f*_2_ = 1 (*f*_1_ + *f*_2_ = 2). 

During the analysis of thermal phenomena, perfect contact between the joined elements is considered. In the case of the analysis of mechanical phenomena, “self-contact” [40] between the inner plane of pipe with a larger diameter and the outer plane of pipe with a smaller diameter is assumed. Figure 15 shows the location of the self-contact. 

Figure 15 shows also the location and type of assumed mechanical boundary conditions. At the end of the smaller diameter pipe, all degrees of freedom were disabled (translations *U_X_* = *U_Y_* = *U_Z_* = 0 and rotations *U_RX_* = *U_RY_* = *U_RZ_* = 0). 

## 5. Results and Discussion

The computer simulations were carried out on the basis of developed mathematical and numerical models. The calculations were performed in two steps. The first step concerns the thermal and mechanical phenomena related to the process of the circumferential welding of pipes of various diameters. In the second step, numerical calculations of mechanical phenomena during the compression test were carried out. The simulation of compression of the pipe was made taking into account the residual stresses resulting from the welding process. 

### 5.1. Results of Numerical Simulations of Thermomechanical Phenomena during Welding Processes

The time consumed by the welding torch to shift along the outer plane of the pipe depends on the radius and the linear heat source travel speed. For accepted data, this time is approx. 20 s. The total simulation time from the start of heating to cooling down to ambient temperature is 150 s. The temperature in the joint is determined for accepted energetic parameters of the heat source. Figure 16 shows the temperature field for two different simulation durations (*t =* 7 s; *t* = 17 s). The melted zone is marked in this figure by solid line, determined by liquidus temperature (*T_L_*). 

Figure 17a shows the temperature field in the cross-section of the joint for time *t* = 10 s. The solid line marks the boundary of the melted zone. The results of the simulation of temperature distribution were verified experimentally. Figure 17b shows a cross-section of the real weld where the boundary of the melted zone (the solid line marks the range of the melted zone, *T*_L_ ≈ 1455 °C) is given in the frame. As can be observed, for assumptions accepted in the model, a good agreement of the results is obtained. 

The numerically determined thermal load acting on the combined system of two pipes of different diameters allowed performing calculations of mechanical phenomena. Figure 18 shows the distribution of temporary reduced stress for two different simulation times, *t* = 8 s and *t* = 17 s. The largest temporary stresses occur in the area of heating source action on the material with the amount of 255 MPa. Residual reduced stress after cooling to ambient temperature is presented in Figure 19 for time *t* = 150 s. 

From the comparison of Figure 18 and Figure 19, it can be seen that after lowering the joint temperature to the ambient temperature (t = 150 s), the maximum stresses decreased.

The displacement field of the welded joint is numerically predicted. Figure 20 shows the displacement field in a general view (*U_z_, U_y_, U_z_*). For the analyzed welded system of two pipes, the highest value of displacements is approx. 0.03 cm (0.3 mm). 

For the assumed mechanical boundary conditions, the highest values of displacements are directed along the *z* axis (*U_z_*, Figure 20c) in the analyzed joint. On the basis of results shown in Figure 20a,b, it can be seen that the cross-section of the joint is deformed due to the operation of the welding source. The displacement values in the direction of the *x* and *y* axes are comparable with the approximate amount of 0.2 mm. 

The analyzed method of welding of pipes can be used for various types of construction support. Therefore, it is necessary to carry out a strength analysis such as a compression test to define mechanical properties of this type of welded construction. 

### 5.2. Results of Numerical Simulations of Compression Tests 

In this work, a numerical model of the compression test was developed on the basis of experimental tests performed using the universal materials testing machine and image correlation system (see Section 3.1). The strength analysis of the joined pipes was carried out in the numerical simulations. Adopted in the calculations, a discrete model is shown in Figure 14.

The sample was loaded by compression forces as shown in the load diagram (Figure 6). The boundary conditions were assumed in accordance with the existing conditions during the experiment. Boundary conditions assumed in the analysis are shown in Figure 21.

All translational and rotational degrees of freedom are received on the lower surface of the sample, at the point of contact of the sample with the lower stationary compression base: *U_x_ = U_y_ = U_z_* = 0, *U_Rx_* = *U_Ry_* = *U_Rz_* = 0 and *U_Rx_* = *U_Ry_* = *U_Rz_* = 0. On the other hand, on the upper contact surface with the movable crossbeam, motion along the *z* axis is allowed (*U_x_* = *U_y_* = 0 and *U_Rx_* = *U_Ry_* = *U_Rz_* = 0). The calculations are carried out in the elastic-plastic range. Material data in the plastic range for X5CrNi18-10 steel are assumed with respect to literature data [48].

Numerical calculations were carried out taking into account the residual stresses generated in the welding process. The numerically estimated residual stress field (Figure 19) was implemented in the calculations as the initial condition. Stress distribution resulting from the axial compression of the sample was numerically estimated (Figure 22) as well as the displacement distribution (Figure 23). 

Figure 22a shows the distribution of reduced stresses in the first second of the simulation (*t* = 1 s). On the other hand, Figure 22b shows the stress distribution for the maximum compressive force load on the sample (*t* = 37.5 s). 

From the analysis of the prediction of stress state in the compressed sample (Figure 22), it can be seen that in the initial stage of loading, the highest stress concentration is located in the welded joint, which is related to the residual stresses from the welding process. The reduced stresses increase in the entire element during loading with the compressive force. At maximum load, the maximum stress values occur in the weld zone (Figure 22b). At the given load of 90 kN, the element is not damaged. However, a further increase in the load could lead to the destruction of the specimen due to a crack in the weld. Figure 23 shows the displacement distributions taking into account the reduced residual stresses resulting from welding process. 

The assumed parameters of the compression test are the same as parameters obtained in the experiment (Figure 6). Therefore, values of longitudinal displacements *U_z_* are consistent with the experiment. The maximum value of this displacement is up to 1.6 mm (the check point refers to the value of the displacement from the testing machine for the test time of 37.5 s). The negative sign of the displacement along the *z* axis direction results from the adopted coordinate system. This direction of the *z* axis is opposite to the direction in the coordinate system found in experimental studies. The obtained displacements in the pipe axis are comparable; their maximum values oscillate around 0.2 mm. Numerically estimated values of displacements in the *x* axis and *y* axis are comparable with results of the experiment obtained by the Dantec 3D system (Figure 9). Figure 23d shows a comparison of displacements along the *z* axis for the measurement line (Figure 23c) with the results obtained in the Dantec 3D system (Figure 8). A high convergence of the obtained results can be observed in this figure.

## 6. Conclusions

Contrary to most of the papers on modeling the welding process, this work simulates the thermomechanical phenomena in welding and tests performed to determine the mechanical properties of the welded construction. Numerical analysis of the welding process as well as the simulation of mechanical loads during compression tests allowed the prediction the strength of the joint.

The comparison of numerically estimated liquidus temperature with the results of the experiment shows that the estimated fusion zone well agrees with the boundary of the melted zone in the cross-section of the obtained joint. The reduced welding stresses obtained in the simulations do not exceed the value of 250 MPa and the displacement values do not exceed 0.3 mm. The obtained simulation results of the welding process are used as input data for the strength analysis during the compression test of welded pipes.

The experimental results obtained from the Dantec 3D image correlation system are consistent both in the transverse direction and along the *z* axis. The comparison of displacement fields in each direction confirmed the correctness of the developed numerical models. 

On the basis of the obtained results of simulation of stresses in the compressed sample (see Figure 22), it can be seen that in the initial stage of loading, the highest stress concentration is located in the joint, which is related to the residual welding stresses. During loading, the residual stress in the entire sample increases. The maximum stress values occur in the weld zone at a maximum load of 90 kN. A further increase in the load could lead to the failure of the sample.

The developed numerical model allows the prediction of the influence of welding parameters on the distribution of stresses and strains, and thus allows the determination of process parameters used to obtain a good quality of the joint. 

## Figures and Tables

**Figure 1 materials-15-03243-f001:**
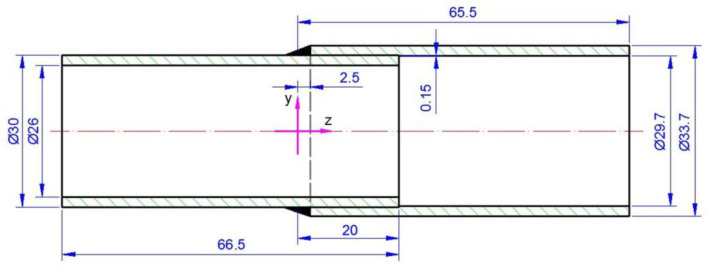
Diagram of the analyzed system.

**Figure 2 materials-15-03243-f002:**
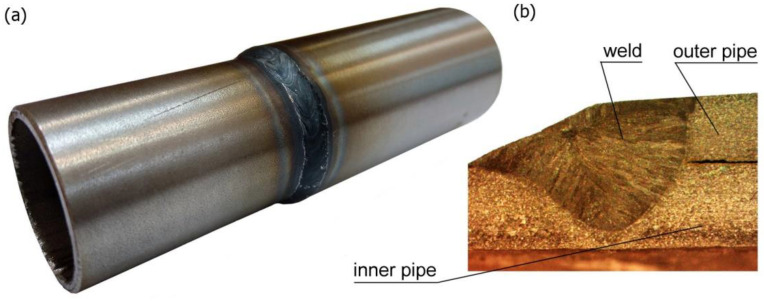
A circumferentially welded pipe of different diameters: (**a**) general view, (**b**) macroscopic view of the cross-section of the weld.

**Figure 3 materials-15-03243-f003:**
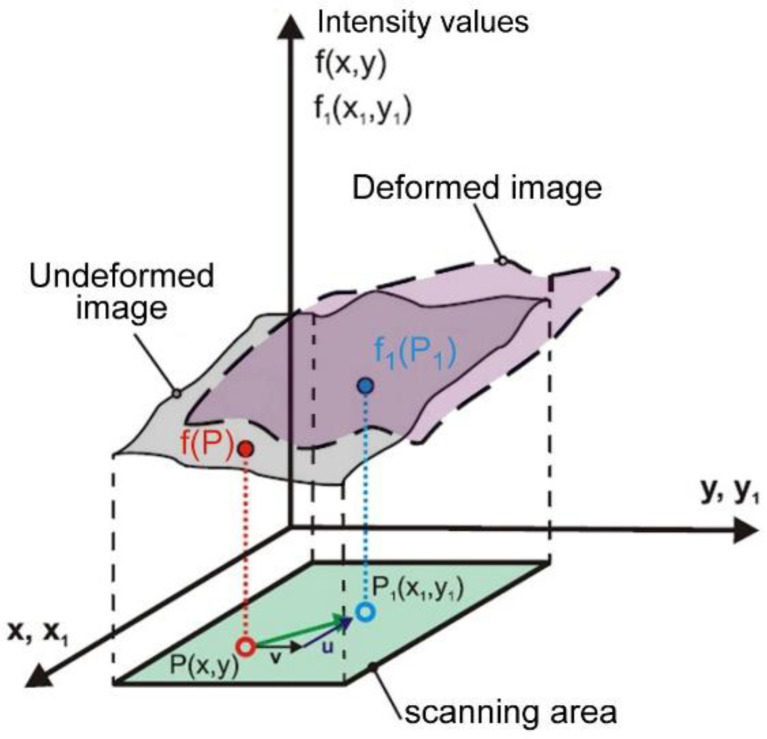
Diagram of surface image analysis before and after deformation [34].

**Figure 4 materials-15-03243-f004:**
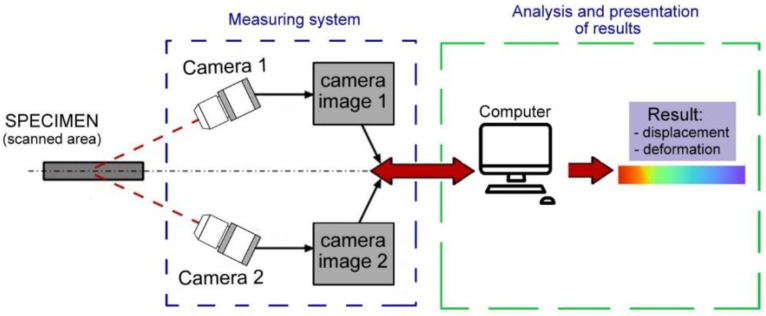
Diagram of deformation measuring system.

**Figure 5 materials-15-03243-f005:**
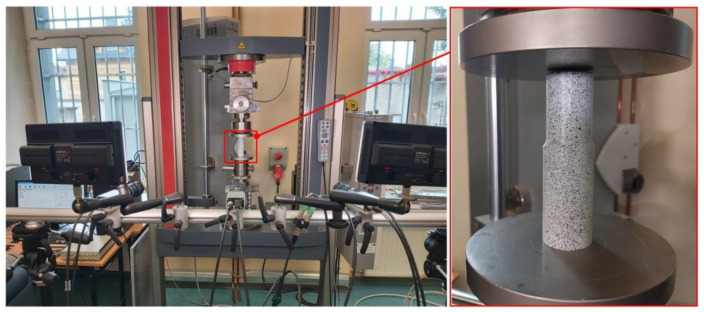
Measurement system for determination of displacements with working area and sample.

**Figure 6 materials-15-03243-f006:**
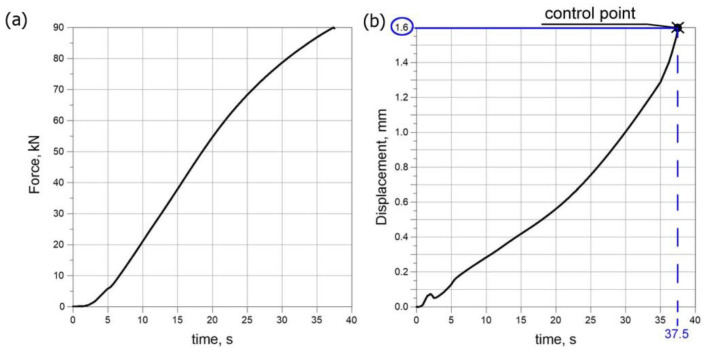
Characteristic parameters of the compression test, (**a**) load diagram, (**b**) displacement diagram of the upper traverse.

**Figure 7 materials-15-03243-f007:**
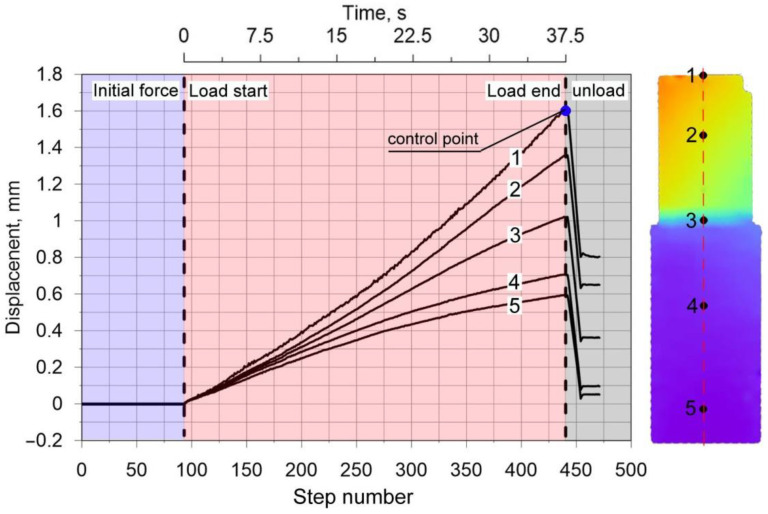
*U_z_* displacement diagram for five selected measuring points.

**Figure 8 materials-15-03243-f008:**
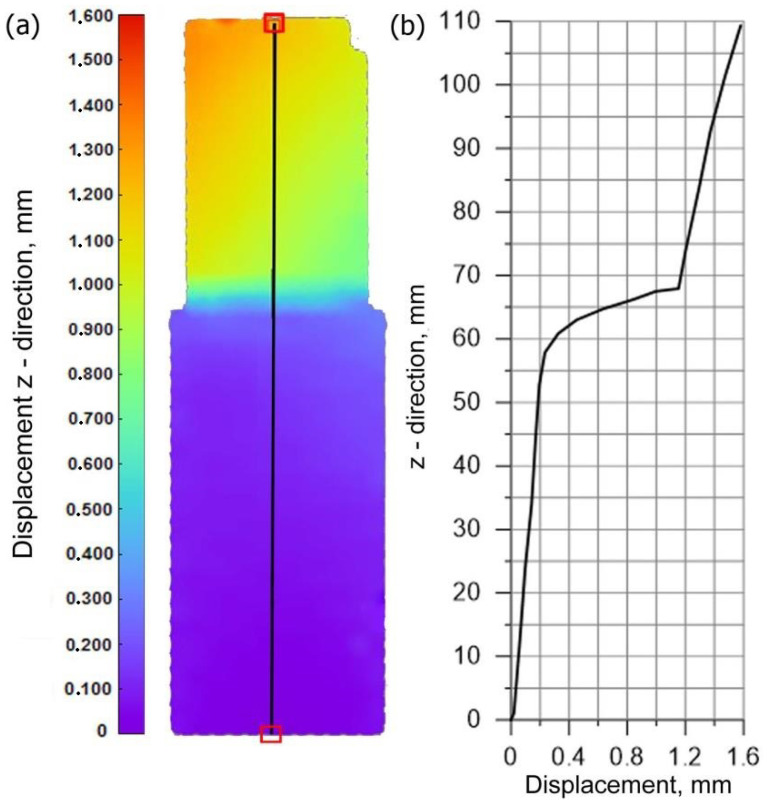
Experimental displacement distribution in the z axis (**a**) on the outer surface, (**b**) along the center lines.

**Figure 9 materials-15-03243-f009:**
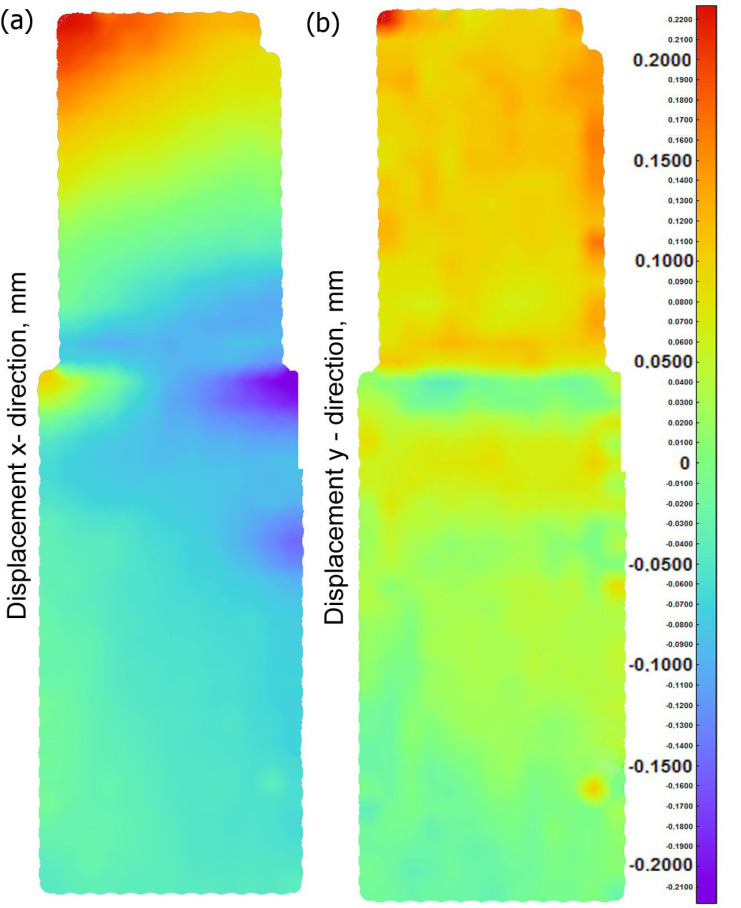
Displacement (**a**) in the x axis, (**b**) in the y axis.

**Figure 10 materials-15-03243-f010:**
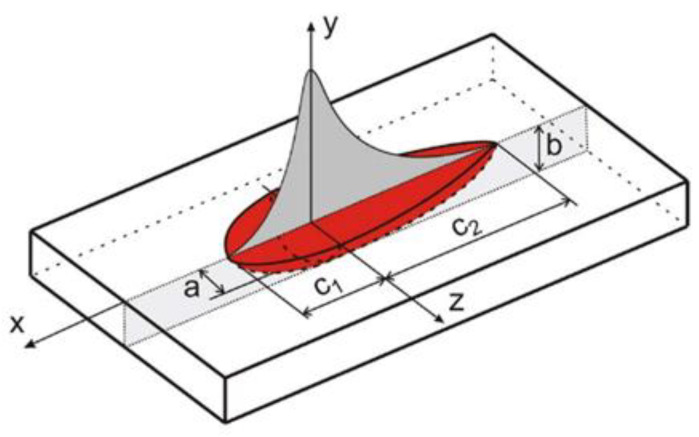
Scheme of Goldak’s model (Equation (13)).

**Figure 11 materials-15-03243-f011:**
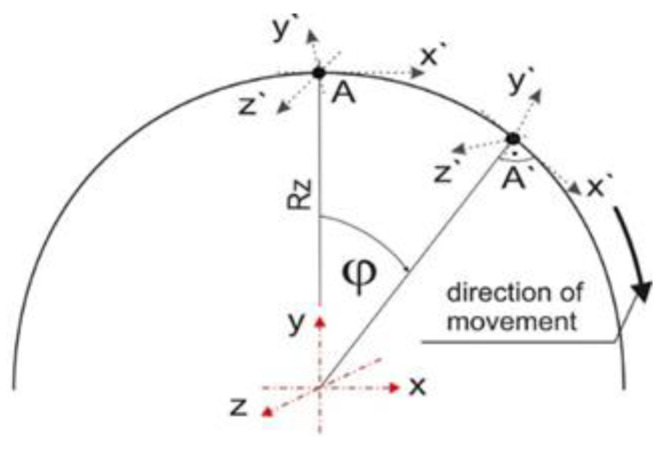
Scheme of the source movement in a cylindrical system [33].

**Figure 12 materials-15-03243-f012:**
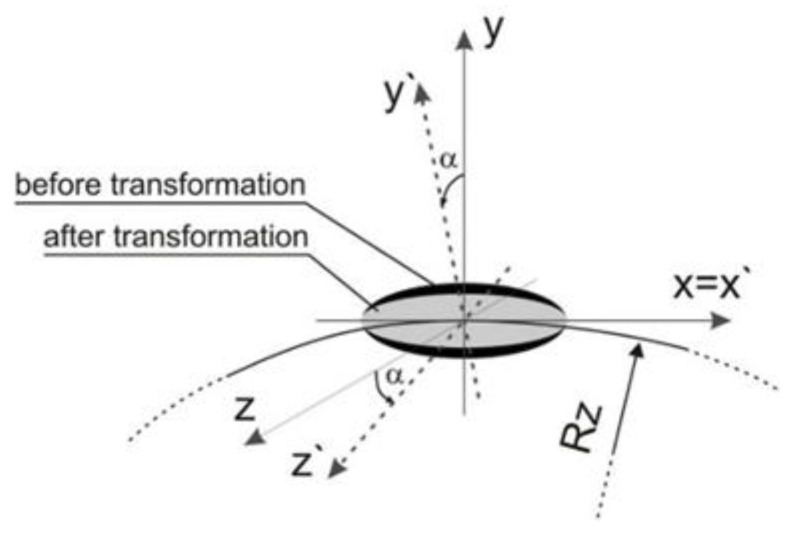
Scheme of transformation system of the Goldak source power distribution [33].

**Figure 13 materials-15-03243-f013:**
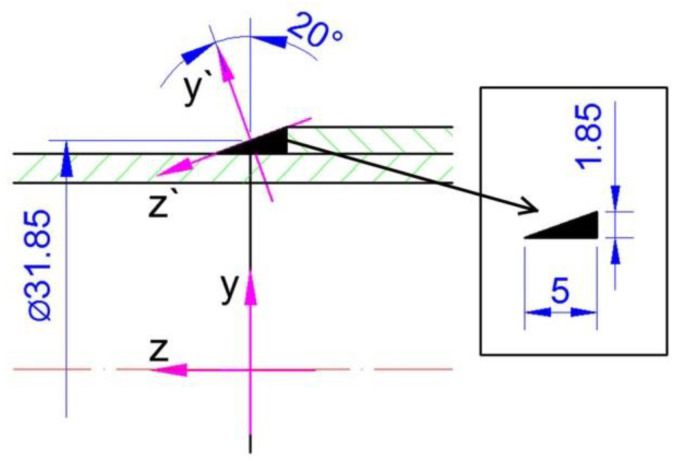
Scheme of the inclination of the heat source.

**Figure 14 materials-15-03243-f014:**
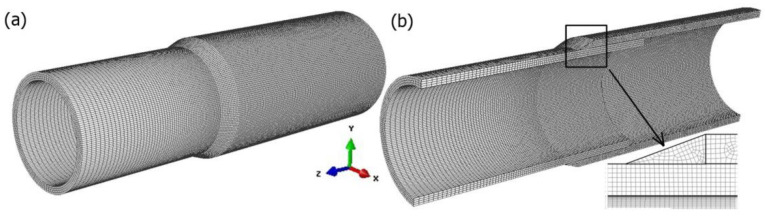
Discrete model of the analyzed joint.

**Figure 15 materials-15-03243-f015:**
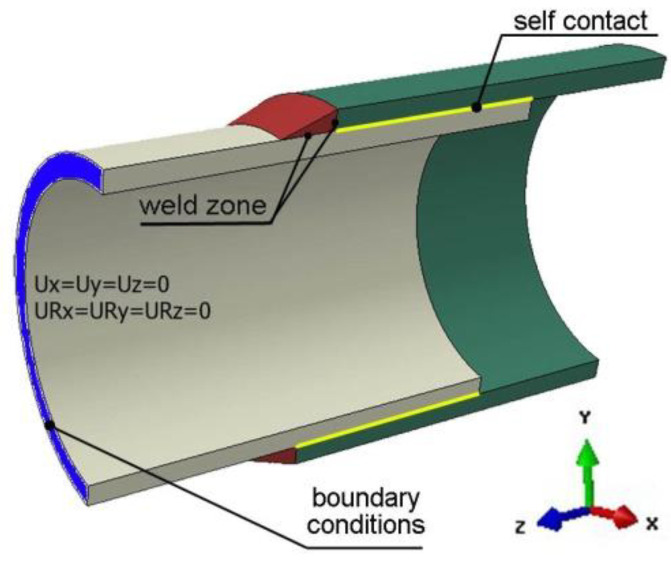
Scheme of contact area and mechanical boundary conditions.

**Figure 16 materials-15-03243-f016:**
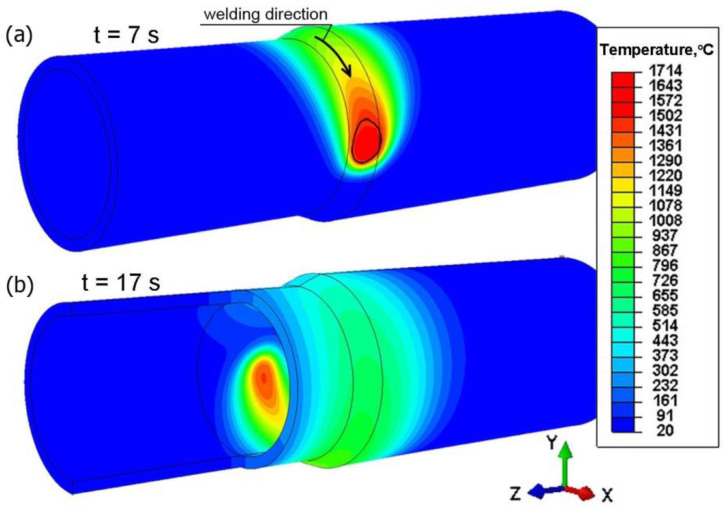
Temperature profile of welded pipes for two simulation times, *t* = 7 s; *t* = 17 s.

**Figure 17 materials-15-03243-f017:**
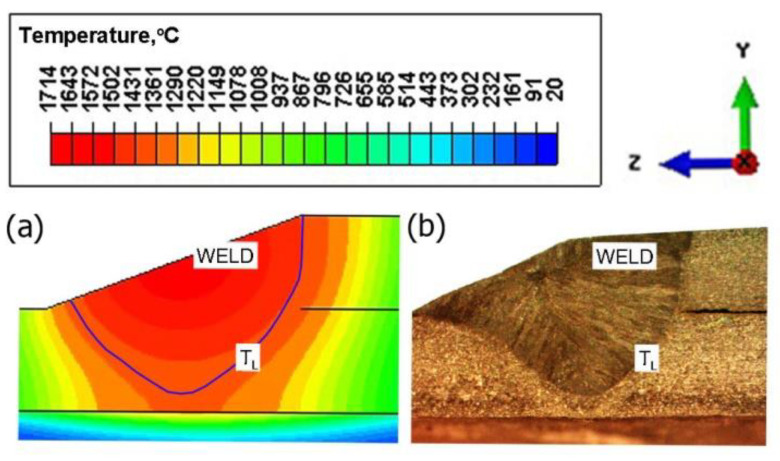
Cross-section of the joint (**a**) results of numerical calculations, (**b**) real weld.

**Figure 18 materials-15-03243-f018:**
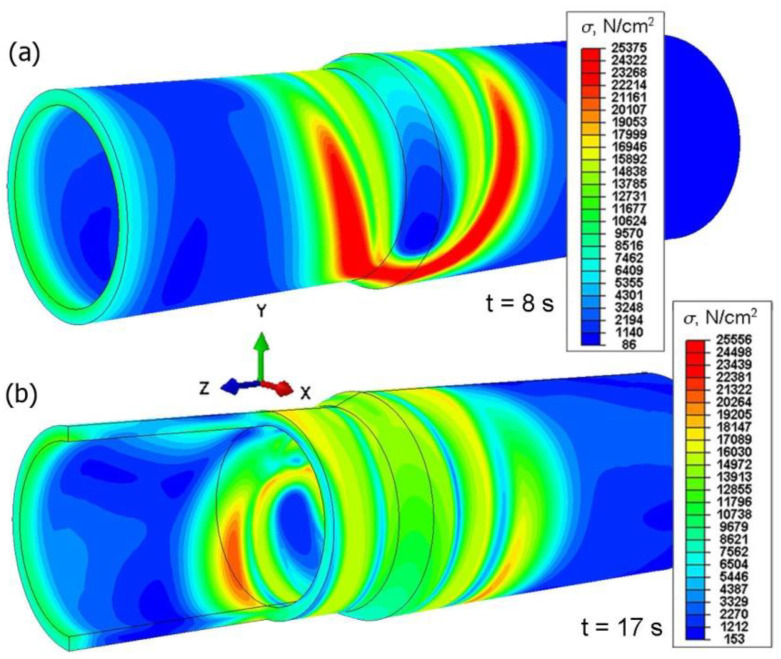
Temporary reduced stresses of a welded joint for two different simulation times, *t* = 8 s; *t* = 17 s.

**Figure 19 materials-15-03243-f019:**
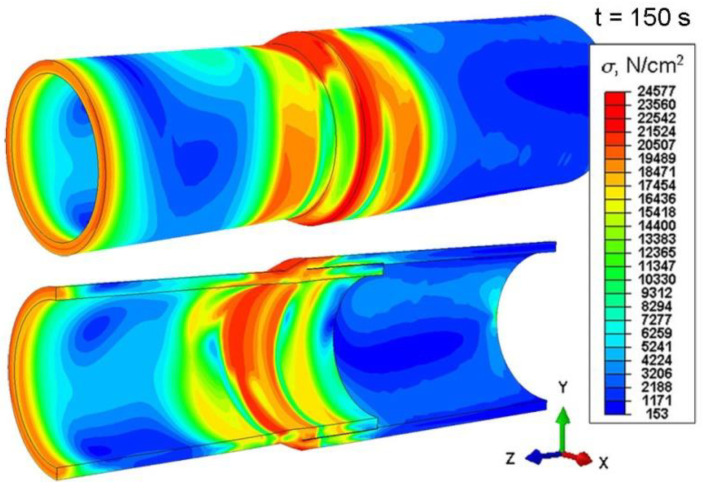
Reduced stresses after welding, for time *t* = 150 s.

**Figure 20 materials-15-03243-f020:**
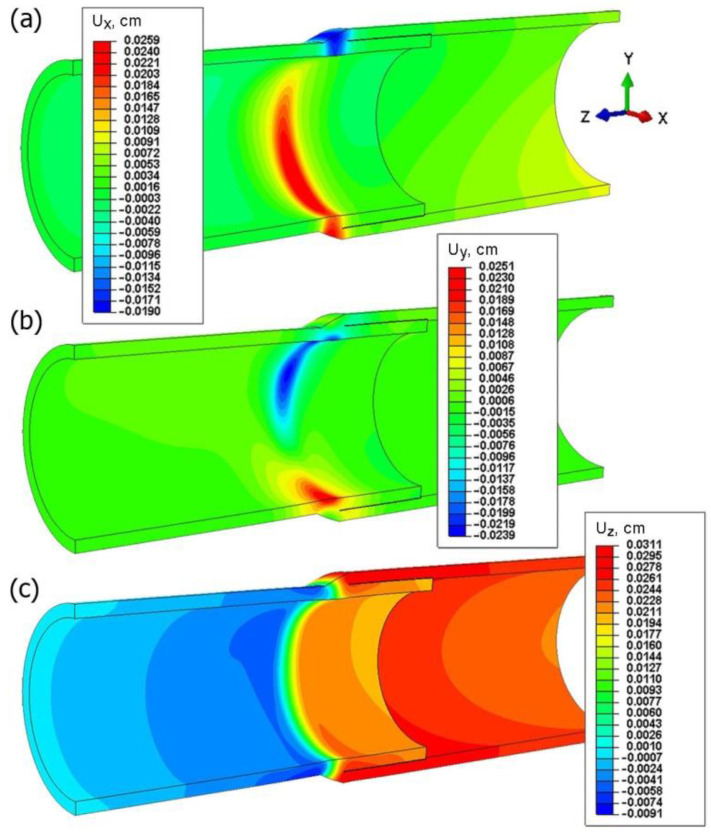
Displacement field in welded pipes.

**Figure 21 materials-15-03243-f021:**
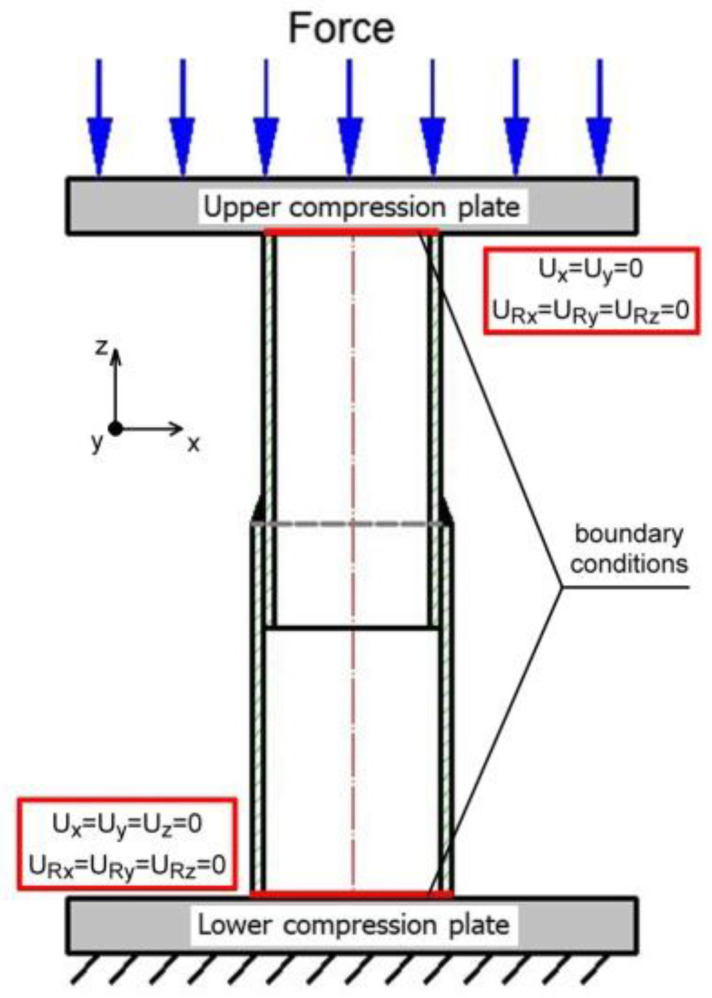
Diagram of the analyzed system during compression test.

**Figure 22 materials-15-03243-f022:**
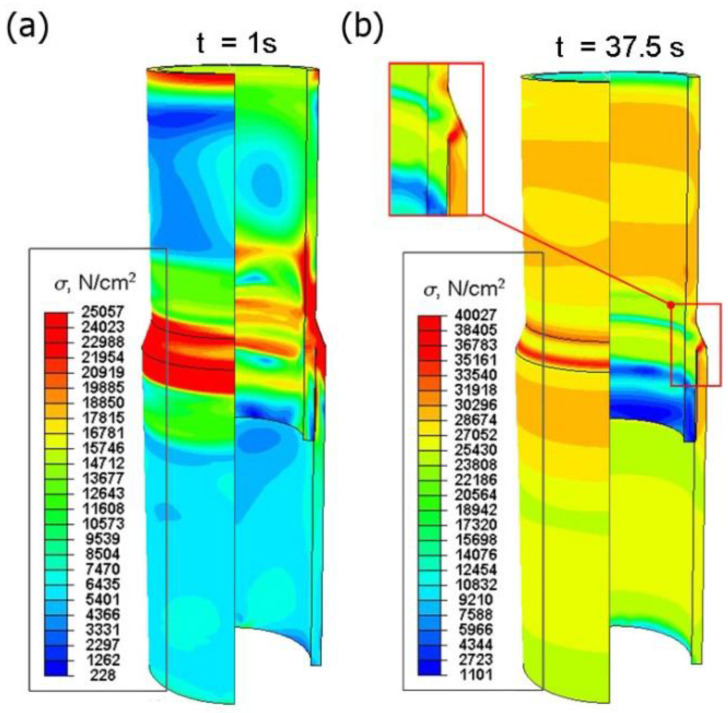
Distribution of reduced stresses in the compressed sample for time (**a**) *t* = 1 s, (**b**) *t* = 37.5 s.

**Figure 23 materials-15-03243-f023:**
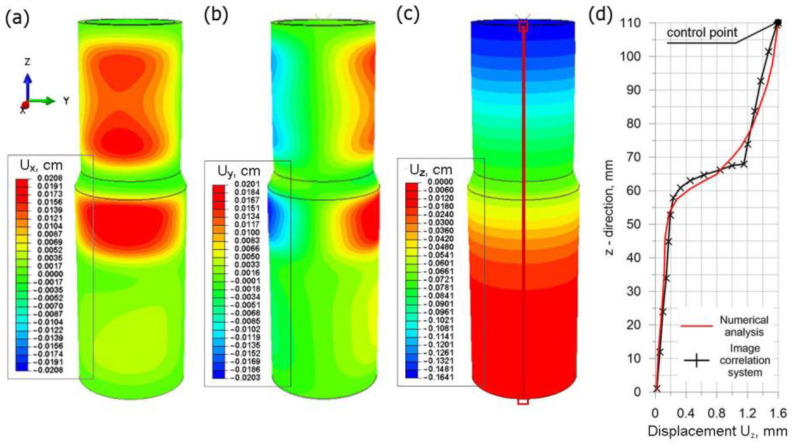
Displacement fields in the compressed sample: (**a**) *U_x_*, (**b**) *U_y_*, (**c**) *U_z_* and (**d**) comparison of *U_z_* displacement with experimental test.

## Data Availability

Data is contained within the article.

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
