# Peer review of "Numerical Prediction of Strength of Socket Welded Pipes Taking into Account Computer Simulated Welding Stresses and Deformations"

_materials, 2022, doi:10.3390/ma15093243_

Round 1

Reviewer 1 Report

The work deals with the simulation and numerical of welding deformation and residual stresses. Due to the increased need to computationally estimate the capacity of structural components, welding simulations have been increasingly applied to understand the performance of welded joints. In general, the topic in the paper is worth investigation and there is novelty regarding the joint type, in the reviewer's opinion, but the paper lack of experimental support for the verification of the simulation model. The experiments have been carried out on the compressive tests that have been simulated, too, but these results do not support the welding simulation model. The only experimental verification on the model is the cross-sectional macrography. Due to these reasons, the reviewer does not find the paper very sound and solid, in terms of scientific readiness, unfortunately. Please find the detailed comments below:

(1) Introduction is well-written, no needs for changes. Great to see well-introduced background on the study.

(2) Section 3 is totally redundant. No need to explain how digital image correlation system works as this paper is not related to the development of DIC systems, it is just an experimental method/equipment used in the study.

(3) Goldak's heat source shape was assumed. The authors could have been using the shape of end pool at the run-off position to estimate a,b,c1 and c2 geometry parameters, see e.g. https://doi.org/10.1016/j.jcsr.2021.107088 (open-access)

(4) Did the authors conduct any temperature measurements (thermo-couples, temp. camera, or else) to verify the heat source model? Otherwise, how the authors can build a FE model without any evidence on actual values?

(5) Figure 19 is somewhat redundant, as the final outcome (welding deformation/residual stress) are meaningful, not the temporary conditions.

(6) Did the authors measure...

(a) welding deformations by triangulation or other system?

(b) welding residual stresses by X-ray or other system?

... This data would be necessary (at least deformations) to verify the heat-source model.

(7) it is somehow unclear how the axial compression tests are related to the topic investigated / entitled in this research work (simulation). Of course, it is nice to see the performance of joint(s) under compression load but still this data does not support the numerical work that is the core of the present study.

(8) Conclusions should be revised to address the actual contribution to knowledge. The conclusions about the usability of a single DIC system is not important for a reader.

Due to these major comments, the reviewer considers that the paper should be majorly revised before publication/acceptance

Reviewer 2 Report

The work is focused on numerical modelling of thermal and mechanical phenomena occurring during socket welding of pipes with various diameters. The fusion zone morphology, temperature field, stress as well as displacements during electric arc welding of pipes were predicted. Also, the compression test was performed and simulated to investigate the mechanical performance of the joint. The results are very useful for the community of socked welding of pipes or other part with similar structure.

The work is useful, but the necessity and novelty of the work were not clearly elaborated in the Introduction Section. What is the improvement compared with other similar work?

In Figure 17, direction of welding should be given.

Line 387, “Figure 17” should be “Figure 19”?

The Abstract and Conclusion Section should be rewritten.

For the Abstract Section, the objective, method, results and conclusion of the work should be given. In present form, the content of method is slightly more, but the content about results and conclusion is relatively weak.

For Conclusion Section, the important results and conclusion should be given, not too much background information. For example, in the first paragragh, “The analyzed flange joint allows to reduce the weight of the structure by about 11% while maintaining the proper stiffness of the structure.” This is not the conclusion from the work, so it should not be given in Conclusion Section. Also, the figure information should not be given in this Section.

Reviewer 3 Report

The work present a numerical/experimental model of a welded pipe under compression stress and the simulation of the welding process of the same pipe. 

Although some interesting aspects have been faced by the authors, and generally the chosen method is correct, in the reviewer opinion it is not clear the correlation between the compression test and the welding simulation. In the reviewer opinion, the authors should made two different works or find some correlations (e.g. welding parameters vs. compression resistance) between the two problems. 
In general the paper need an in depth re-organization in way to clarify the aims of work. 

Round 2

Reviewer 1 Report

The authors have revised the manuscript as per the comments given by the reviewer(s). The reviewer understands that some of the truly required measurements for the verification of simulation model are not available, at least afterwards. However, due to these drawbacks, the paper is not very solid to contribute to the knowledge in the field of welding simulations. This can be also regarded as general comment to authors when they consider implementation of their next studies and research paper. Consequently, the reviewer still suggests few changes to consider these shortages in the content and to not mislead a reader, see comments below:

(1) Due to these lacks of experiments and the content of the paper, the reviewer suggest title "Buckling capacity of socket-welded pipe connections considering simulated welding residual stresses and deformation" or equivalent. The main contribution of the paper is the combined simulation and mechanical analysis on the compressed tubes.

(2) The reviewer would hesitate to conclude anything about the verification of simulation model (due to these shortages of exp. data) and would thus remove the second paragraph from the conclusion section ("From the comparison...")

Reviewer 2 Report

The Abstract has been corrected and improved. However, the most was focused on the method and what was done in this work. The content about results and conclusion in this work is still weak.

For the Introduction Section, the necessity and novelty of the work were not clearly elaborated. The improvement compared with other similar work should be emphasized.

For the Section 3 “Image Correlation System”, The reviewer noticed that most content has been deleted. Why did it?

The conclusion Section has been corrected and improved a lot. It is advised to improve it further.
